# Validation of the T-Lymphocyte Subset Index (TLSI) as a Score to Predict Mortality in Unvaccinated Hospitalized COVID-19 Patients

**DOI:** 10.3390/biomedicines10112788

**Published:** 2022-11-02

**Authors:** Andrea Di Lorenzo, Simona Tedde, Pier Giorgio Pace, Laura Campogiani, Lorenzo Ansaldo, Alessandra Lodi, Marta Zordan, Filippo Barreca, Federica Caldara, Benedetta Rossi, Alessandra Imeneo, Grazia Alessio, Angela Maria Antonia Crea, Davide Checchi, Vincenzo Malagnino, Elisabetta Teti, Luigi Coppola, Raffaele Palmieri, Francesco Buccisano, Massimo Andreoni, Loredana Sarmati, Marco Iannetta

**Affiliations:** 1Department of System Medicine, Tor Vergata University, 00133 Rome, Italy; 2Infectious Disease Clinic, Policlinico Tor Vergata, 00133 Rome, Italy; 3Department of Biomedicine and Prevention, Tor Vergata University, 00133 Rome, Italy

**Keywords:** SARS-CoV-2, TBNK, survival, severity, peripheral blood, cytokine, leukocyte

## Abstract

Lymphopenia has been consistently reported as associated with severe coronavirus disease 2019 (COVID-19). Several studies have described a profound decline in all T-cell subtypes in hospitalized patients with severe and critical COVID-19. The aim of this study was to assess the role of T-lymphocyte subset absolute counts measured at ward admission in predicting 30-day mortality in COVID-19 hospitalized patients, validating a new prognostic score, the T-Lymphocyte Subset Index (TLSI, range 0–2), based on the number of T-cell subset (CD4+ and CD8+) absolute counts that are below prespecified cutoffs. These cutoff values derive from a previously published work of our research group at Policlinico Tor Vergata, Rome, Italy: CD3+CD4+ < 369 cells/μL, CD3+CD8+ < 194 cells/μL. In the present single-center retrospective study, T-cell subsets were assessed on admission to the infectious diseases ward. Statistical analysis was performed using JASP (Version 0.16.2. JASP Team, 2022, Amsterdam, The Netherlands) and Prism8 (version 8.2.1. GraphPad Software, San Diego, CA, USA). Clinical and laboratory parameters of 296 adult patients hospitalized because of COVID-19 were analyzed. The overall mortality rate was 22.3% (66/296). Survivors (S) had a statistically significant lower TLSI score compared to non-survivors (NS) (*p* < 0.001). Patients with increasing TLSI scores had proportionally higher rates of 30-day mortality (*p* < 0.0001). In the multivariable logistic analysis, the TLSI was an independent predictor of in-hospital 30-day mortality (OR: 1.893, *p* = 0.003). Survival analysis showed that patients with a TLSI > 0 had an increased risk of death compared to patients with a TLSI = 0 (hazard ratio: 2.83, *p* < 0.0001). The TLSI was confirmed as an early and independent predictor of COVID-19 in-hospital 30-day mortality.

## 1. Introduction

In December 2019, a new type of betacoronavirus was identified in Wuhan, China, although some mathematical models suggest that it might have emerged between early October and mid-November 2019. By January 2020 it had spread globally [1]. This new virus was named Severe Acute Respiratory Syndrome Coronavirus 2 (SARS-CoV-2) because of its similarities to the previously identified Severe Acute Respiratory Syndrome Coronavirus (SARS-CoV). The disease it causes was named coronavirus disease 2019 (COVID-19) [2,3]. SARS-CoV-2 infection may vary from asymptomatic to a severe respiratory illness requiring hospitalization in an intensive care unit (ICU) and mechanical ventilation [4,5]. Several studies have analyzed demographic, clinical, and laboratory parameters associated with increased COVID-19 severity. Besides age, comorbidities, and male sex, the reduction in circulating lymphocyte subsets seems to play a crucial role in the evolution of COVID-19 to the most severe clinical forms [5]. Indeed, lymphopenia and immune system dysregulation have been consistently reported as associated with more serious manifestations of COVID-19 [5,6]. Several studies further investigating COVID-19–associated lymphopenia described a profound decline in all T-cell subtypes [7] in COVID-19 hospitalized patients with severe and critical COVID-19 and an inverse correlation between CD8+ and CD4+ T-cell lymphopenia and the degree of inflammation observed in these patients’ subset. [6,8]. Conversely, a progressive T-cell count normalization has been described in mild COVID-19 cases and in recovering patients [9]. Monitoring T-cell absolute count variations during SARS-CoV-2 infection could serve as a prognostic marker of increased risk of mortality [10,11]. The analysis of immunologic impairment evolution over time could help to predict COVID-19 disease progression; likewise, the T-cell count measured at COVID-19 onset might have prognostic value. Given the crucial role of immunologic impairment in COVID-19 disease progression, lymphopenia has been proposed and evaluated as an early prognostic marker for SARS-CoV-2–infected individuals. In a previous study from our group, we identified definite cutoff values for T-lymphocyte subset absolute counts (CD3+, CD3+CD4+, and CD3+C8+) assessed at ward admission and associated with an increased risk of in-hospital mortality for COVID-19 patients [6].

The aim of this study was to evaluate the role of T-lymphocyte subset absolute counts assessed at ward admission in predicting 30-day mortality in COVID-19 hospitalized patients; we accomplish this by validating previously obtained cutoffs for total CD3+, CD4+, and CD8+ T-cell absolute counts and by applying a new prognostic score based on the number of T-cell subset absolute counts below these cutoffs.

## 2. Materials and Methods

### 2.1. Patients, Laboratory Parameters, and Data Collection

This is a single-center retrospective study, involving patients hospitalized because of COVID-19 in the infectious disease (ID) ward of Policlinico Tor Vergata University Hospital in Rome, Italy, from 2 September to 31 December 2020.

We included adult patients (>18 years old), with at least one positive reverse transcription polymerase chain reaction (RT-PCR) test for the detection of SARS-CoV-2 RNA on a nasopharyngeal (NPh) swab. Only COVID-19 hospitalized patients with peripheral blood (PB) lymphocyte subset absolute counts assessment at ward admission were included in the study. Patients were classified as non-survivors (NS) if death occurred within 30 days of hospitalization. Patients were classified as survivors (S) who were either: (1) discharged home, (2) moved to a residential structure for SARS-CoV-2–positive patients, or (3) moved to a different ward of the hospital and still alive at the 30th day from hospital admission. Patients for whom a definite outcome could not be retrieved were excluded from the study.

We also classified patients in five groups on the basis of the maximal oxygen supply required during hospitalization: ambient air (AA), Venturi oxygen mask (VMK), nonrebreather oxygen mask with concentrator (NRM), noninvasive ventilation (NIV), and invasive mechanical ventilation through orotracheal intubation (OTI).

Blood tests were performed in the hospital central laboratory according to standard procedures. PB T-(total CD3+, CD3+CD4+, CD3+CD8+) B-(CD19+) and Natural Killer (NK)-(CD3negCD16+CD56+) lymphocyte absolute counts were assessed with multiparametric flow cytometry according to a standardized procedure. A more detailed description of the assay can be found in Iannetta et al. [6].

T-cell subset absolute counts were compared with cutoffs previously published by our group [6] (CD3+ < 524 cells/μL, CD3+CD4+ < 369 cells/μL, CD3+CD8+ < 194 cells/μL), and the “T-lymphocyte subset index” (TLSI) was calculated for each of the enrolled patients, defined as the number of T-lymphocyte CD3+CD4+ and CD3+CD8+ absolute counts below the cutoff value, ranging from 0 to 2.

An ad hoc electronic database was created to collect demographic and clinical data, laboratory parameters, and T-B-NK-lymphocyte subset absolute counts.

The study was approved by the Ethics Committee of Fondazione PTV Policlinico Tor Vergata (register number 154/21). The requirement for patients’ informed consent was waived by the Ethics Committee, considering the retrospective nature of the study, in accordance with local legislation.

### 2.2. Statistical Analysis

Quantitative data are shown as median and interquartile range (IQR); qualitative data are shown as absolute and relative (percentage) frequency. Differences between groups were assessed using the Mann–Whitney U test, the Kruskal–Wallis test (continuous variable), or the Chi^2^ test (categorical variables), as appropriate.

The Spearman’s correlation test was used to analyze linear correlation. Univariable and multivariable regression analyses were performed. We considered a two-sided *p* value < 0.05 as statistically significant. Statistics were performed using JASP (Version 0.16.2. JASP Team, 2022, Amsterdam, The Netherlands) and Prism 8 for macOS (version 8.2.1. GraphPad Software, San Diego, CA, USA).

## 3. Results

### 3.1. Study Population

Three-hundred-forty patients were screened from 2 September to 31 December 2020: 44 patients were excluded due to missing baseline information, reaching a final study cohort of 296 consecutive patients hospitalized in the ID ward of Policlinico Tor Vergata Hospital, Rome, Italy, because of COVID-19. None of the enrolled patients had received SARS-CoV-2 vaccination. The median age of the enrolled population was 67 years [IQR 55–76], with a prevalence of males (70.1%). The median time from the onset of the symptoms to the first positive SARS-CoV-2 NPh swab was 7 days [IQR 3–10], and the median time from the onset of the symptoms to the lymphocyte subsets assessment was 9 days [IQR 5–12] (Table 1).

The majority of the patients needed oxygen/ventilation support: 74 patients (25.0%) with VMK, 16 (5.4%) with NRM, 99 (33.4%) with NIV, and 48 (16.2%) with OTI, while 59 patients (20.0%) did not receive any oxygen/ventilation support (Table 1).

As for comorbidities, 84.4% of the enrolled subjects had at least one, and the median number of comorbidities was 2 [IQR 1–3]. Specifically, cardiovascular diseases were the most common concomitant clinical condition, accounting for 62.6% of cases. The overall 30-day mortality rate was 22.3%. Comparing non-survivors (NS) and survivors (S), NS were older and prevalently males compared to S (*p* < 0.001 and *p* = 0.044, respectively) (Table 1). Cardiovascular, cerebrovascular, pulmonary, renal, and hematological diseases and solid organ tumors were significantly more common in NS compared to S (Table 1).

The median time from the onset of the symptoms to the first SARS-CoV-2 NPh swab and to the first PB lymphocytes assessment did not significantly differ between NS and S (5 vs. 7 days, *p* = 0.075 and 8 vs. 9 days, *p* = 0.089, respectively) (Table 1).

### 3.2. Laboratory Findings

In the whole cohort, C-Reactive protein (CRP) D-Dimers and fibrinogen were above normality range, according to the hospital laboratory reference values (Table 2).

A pro-inflammatory state characterized by significantly higher values of white blood cell (WBC), neutrophil (N) absolute count, lactic dehydrogenase (LDH), IL-6, D-dimers, and CRP was observed in NS compared to S patients (*p* = 0.004; *p* < 0.001; *p* < 0.001; *p* < 0.001; *p* < 0.001; *p* < 0.001; *p* < 0.001, respectively) (Table 2). A significantly lower lymphocyte (L) absolute count and therefore a higher neutrophil-to-lymphocyte (N/L) ratio were observed in the NS compared to the S group (*p* < 0.001 and *p* < 0.001, respectively) (Table 2).

As for PB lymphocyte subset absolute counts, median CD3+, CD3+CD4+, and CD3+CD8+ absolute counts were reduced in the whole cohort. CD19+ and CD3negCD16+CD56+ absolute counts as well as the CD4/CD8 ratio were within laboratory normality ranges (Table 3). Non-survivors had significantly lower absolute counts of total CD3+, CD3+CD4+, CD3+CD8+ and CD19+ compared to the S group (*p* < 0.001, *p* < 0.001, *p* < 0.001 and *p* < 0.001, respectively) (Table 3).

When comparing median lymphocyte absolute count to our previously identified cutoffs, 86/230 patients (37.4%) in the S group and 49/66 patients (74.2%) in the NS group had a CD3+ absolute count below the cutoff (*p* < 0.001, cutoff CD3+ < 524 cells/μL). Accordingly, a significantly higher percentage of NS had CD3+CD4+ and CD3+CD8+ T-lymphocyte subset absolute counts below the prespecified cutoffs compared to survivors: 109/230 patients (47.4%) in the S group and 52/66 patients (78.8%) in the NS group had a CD3+CD4+ absolute count below the cutoff (*p* < 0.001; cutoff, CD3+CD4+ < 369 cells/μL); 106/230 patients (46.1%) in the S group and 54/66 patients (81.8%) in the NS group had a CD3+CD8+ absolute count below the cutoff (<0.001; cutoff, CD3+CD8+ < 194 cells/μL).

As for the T-lymphocyte subset index (TLSI—number of T-lymphocyte subset absolute counts below the cutoff value, range 0–2), in the overall cohort the median value was 1.00 [IQR 0.00–2.00]; in the S group, 91 (39.6%), 63 (27.4%), and 76 (33.0%) patients out of 230 had a TLSI of 0, 1, and 2, respectively. Conversely, in the NS group, 6 (9.1%), 14 (21.2%) and 46 (69.7%) patients out of 66 had a TLSI of 0, 1, and 2, respectively, with a statistically significant difference between the two groups (*p* < 0.0001). A higher TLSI score was associated with an increased mortality rate, with a trend proceeding from 0 through 1 to 2 (*p* < 0.0001) (Figure 1).

Furthermore, the TLSI was positively correlated with N/L ratio (Spearman rho 0.68 and *p* < 0.001), CRP (Spearman rho 0.37 and *p* < 0.001), LDH (Spearman rho 0.41 and *p* < 0.001), D-dimer (Spearman rho 0,22 and *p* < 0.001), conjugated bilirubin (Spearman rho 0.143 and *p* = 0.017), creatinine (Spearman rho 0.11 and *p* = 0.062), fibrinogen (Spearman rho 0.284 and *p* < 0.001), and international normalized ratio (INR) (Spearman rho 0.13 and *p* = 0.045). IL-6, ferritin, creatinine, and CPK were not significantly correlated with the TLSI.

### 3.3. Multiple Logistic Regression Analysis

At the multiple logistic regression analysis, after adjusting for age, sex, days from the onset of the symptoms to the PB lymphocyte subsets assessment, number of comorbidities, serum IL-6, and LDH, the TLSI emerged as an independent predictor of 30-day mortality, with an odds ratio of 1.89 (adjusted *p* = 0.033) (Table 4).

Similar results were obtained considering the composite endpoint ICU-admission/30-day mortality, confirming the role of the TLSI as an independent predictor of either ICU admission or in-hospital mortality in COVID-19 patients (Appendix A).

### 3.4. Survival Curves

A survival analysis was performed after stratifying COVID-19 patients according to the TLSI score in TLSI = 0 and >0. COVID-19 patients with a TLSI > 0 had an increased risk of death, with a hazard ratio of 2.83 compared to COVID-19 patients with TLSI = 0 (Log-rank Mantel-Cox test *p* < 0.001). Considering death within 30 days from hospitalization, 6/97 (6.2%) and 60/139 (30.2%) patients died within the TLSI = 0 and TLSI > 1 group, respectively (*p* < 0.001). The COVID-19 patients were further stratified into three groups: TLSI = 0, 1, and 2. A statistically significant trend of increasing mortality was observed in COVID-19 patients with a TLSI = 0 through TLSI = 1 to TLSI = 2 (Log-rank Mantel-Cox test *p* < 0.0001; log-rank test for trend *p* < 0.0001), with 6/97 (6.2%), 14/77 (18.2%) and 46/122 (37.7%) death events for COVID-19 patients with a TLSI = 0, 1, and 2, respectively (Figure 2).

Similar results were obtained after performing a time-to-event analysis considering the composite endpoint of ICU-admission/30-day mortality: COVID-19 patients with a TLSI > 0 had an increased risk (hazard ratio 2.75, Log-rank Mantel-Cox test *p* < 0.0001) of either ICU admission or 30-day mortality compared to patients with a TLSI = 0 (Appendix A).

## 4. Discussion

We analyzed the clinical and laboratory parameters of 296 hospitalized adult patients with confirmed SARS-CoV-2 infection to validate the TLSI as an early predictor of COVID-19 in-hospital mortality.

In this study, as widely demonstrated in the literature, older age, the presence of at least one comorbidity, and male sex were confirmed as risk factors associated with increased COVID-19 mortality [12,13]. In our cohort, among the collected comorbidities, cardiovascular, cerebrovascular, chronic pulmonary, renal, and hematologic diseases and solid organ tumors were significantly associated with increased mortality. In contrast with part of the current literature, in our cohort obesity was not associated with increased mortality [14]. Obesity and other metabolic disorders have been associated with severe forms of COVID-19 [15]. Chen R. et al. showed that obese patients had increased mortality rates only in the group of severe COVID-19 patients [9]. In our cohort, both severe and non-severe patients were included, which can justify the lack of a statistically significant association between obesity and increased mortality in the overall cohort. Moreover, in the present study obesity was recorded as a categorical variable (considering a body mass index [BMI] cutoff of 30 kg/m^2^) without recording the exact BMI value for each patient. This could represent a limitation of our analysis.

In COVID-19 disease, a pro-inflammatory state characterized by elevated LDH, IL-6, D-dimers, and CRP concentrations has been frequently reported as a condition predicting a fatal outcome [16]. An elevated leukocyte count driven by increased neutrophils and a concomitant lymphopenia (with high N/L ratio) have also been associated with a worse COVID-19 outcome [17,18]. The present study confirmed these results, showing an association between the alteration of these laboratory parameters and higher mortality. Lymphopenia has been consistently reported as a peculiar finding in SARS-CoV-2 infection, even in non-severe patients, although the most marked reduction has been observed in severe patients and non-survivors [5,19]. The reason for this profound lymphopenia in severe COVID-19 patients is still not fully understood. T-cell loss has been attributed not only to uncontrolled activation and exhaustion but also to defects in IL-2/IL-2 receptor signaling [20]. Besides the reduction in their absolute count, T-cells of severe COVID-19 patients express activation and exhaustion markers and a dysfunctional antiviral adaptive immune response to SARS-CoV-2 [21].

Focusing on T-cell subsets, several studies have highlighted the role of T-lymphocyte subset absolute counts assessed at hospitalization in predicting the risk of death in COVID-19 patients. Xiong L. et al. evaluated lymphocyte subsets in 85 COVID-19 fatal cases and found that an increase in neutrophils absolute count and a decrease in CD4+ T-cell absolute count were independent risk factors for mortality [22]. Overall, in our population, median CD3+, CD3+CD4+, and CD3+CD8+ T-lymphocyte absolute counts were reduced, while CD19+ B-lymphocyte and CD3negCD16+CD56+ NK-cell absolute counts were within laboratory normality ranges. Absolute counts of total CD3+, CD3+CD4+, CD3+CD8+, and CD19+ lymphocytes were significantly lower in non-survivors compared to survivors. Considering the concomitant reduction in both CD4+ and CD8+ T-lymphocyte absolute counts, the CD4/CD8 ratio remained within laboratory normality ranges, as already shown in the literature [12,23].

Considering the cutoffs previously identified by our group [6] in a different cohort, in the present study population a significantly higher percentage of non-survivors had CD3+, CD3+CD4+, and CD3+CD8+ absolute counts below the corresponding cutoffs, compared to survivors. As for CD19+ B-lymphocytes, no differences between survivors and non-survivors were detected in the previous study; hence, no cutoff is available, probably due to the different population sizes. Here we additionally propose the TLSI as an independent predictor of increased mortality in COVID-19 patients, demonstrating good accuracy in the index to reveal patients at higher risk for 30-day mortality, based on T-cell subset absolute counts assessed on hospital admission. The index is obtained by accounting for the number of T-lymphocyte subsets (CD3+CD4+, CD3+CD8+) below the identified cutoffs and can range from 0 to 2. The TLSI was also able to independently predict the composite event of ICU-admission/30-day in-hospital mortality. Furthermore, the TLSI was directly correlated with laboratory markers, such as CRP, LDH, D-dimer, conjugated bilirubin, creatinine, fibrinogen, INR, and N/L ratio, confirming the close connection between hyperinflammation and T-lymphocyte subsets imbalance [16,23,24,25].

Notably, the cutoffs for lymphocyte subset absolute counts identified by our group (CD3+ < 524 cells/μL; CD3+CD4+ < 369 cells/μL; CD3+CD8+ < 194 cells/μL) [6] were similar to those described by other groups in different settings. Pan P et al. found that a value of CD3+ T-cells < 510.5/µL and IL-6 > 6.58 pg/mL were predictors of progression to severe disease in a cohort of COVID-19 patients from China [26]. Belaid B et al. analyzed PB lymphocyte subsets and cytokine levels in 57 COVID-19 patients from North Africa and, using multivariate logistic regression analysis and ROC curves, found that IL-6 > 106.44 pg/mL and CD8+ T-cells < 150/μL were predictive of increased mortality [27]. Lombardi et al. examined the role of CD4+ T-cell absolute count as a predictor of higher mortality in an Italian cohort and found that CD4+ T-cells < 267/μL were associated with higher mortality (sensitivity 82% and specificity 70%) [28]. Xiong L et al. found that a CD4+ T-cell count greater than 260 cells/µL was associated with a reduced risk of death in COVID-19 patients from Wuhan, China. Wang L et al. observed that CD4+ and CD8+ T-cell counts below 405 cells/µL and 182 cells/µL, respectively, could identify COVID-19 patients with increased risk of progression and death in Shanghai, China [29].

The implications of T-lymphocyte subsets in COVID-19 pathogenesis led us to consider the TLSI as extremely useful for risk stratification, allowing for an early identification of patients with an increased risk of death. In the evolving SARS-CoV-2 pandemic, it is crucial to rely on good predictive parameters to promptly assess the risk of progression and death in COVID-19 hospitalized patients [13,23,30,31,32].

Our study has some limitations, such as its retrospective and single-center design. It also has some strengths, namely the relevant number of COVID-19 patients with the baseline assessment of PB lymphocyte subsets included in the study, lager than other similar studies in the literature.

## 5. Conclusions

In conclusion, considering that point-of-care tests for assessing total CD3+ and CD3+CD4+ T-lymphocyte (and by subtraction CD3+CD8+ T-lymphocyte) absolute counts are already available, the TLSI could represent a useful tool for rapid and accurate mortality risk stratification in COVID-19 patients on hospital admission.

## Figures and Tables

**Figure 1 biomedicines-10-02788-f001:**
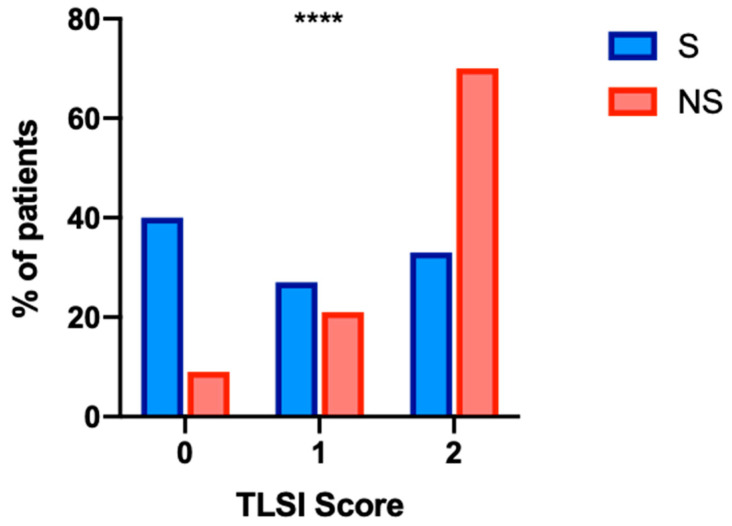
Thirty-day mortality rates (%) in COVID-19 patients according to TLSI score. Survivors (S) are represented in blue, and non-survivors (NS) are represented in red. The analysis was performed with the Chi^2^ test. The Chi^2^ test for trend was also used, showing a statistically significant trend for increasing 30-day mortality rates proceeding from TLSI = 0 to TLSI = 2, *p* < 0.0001. TLSI: T-Lymphocyte subset index; ****: *p* < 0.0001.

**Figure 2 biomedicines-10-02788-f002:**
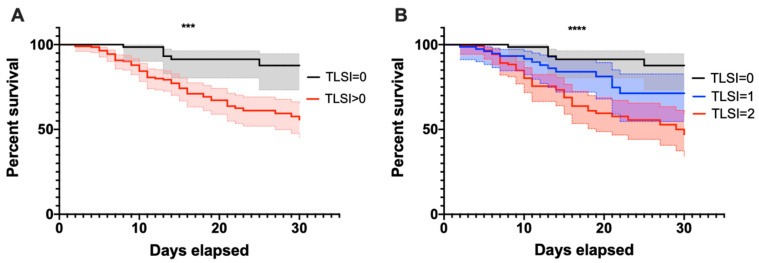
Thirty-day survival curves of COVID-19 patients according to TLSI score. Survival curves at 30 days after the onset of symptoms in hospitalized COVID-19 patients, stratified into two (**A**) and three (**B**) groups, according to TLSI score, ranging from 0 to 2. The analysis was performed with the Log-rank (Mantel-Cox) test. The log-rank test for trend was also used for the survival analysis of the cohort after stratification into three groups (**B**), showing a statistically significant trend from TLSI = 0 to TLSI = 2, *p* < 0.0001. Dashed lines delimit the 95% interval confidence area, which was filled in with the corresponding color. TLSI: T-Lymphocyte subset index; ***: 0.001 < *p* < 0.0001; ****: *p* < 0.0001.

**Table 1 biomedicines-10-02788-t001:** Demographic and clinical characteristics of the study cohort, overall and after stratification for outcome (survivors and non-survivors).

	All Patients (N = 296)	Survivors (N = 230; 77.7%)	Non-Survivors (N = 66; 22.3%)	*p*
Age: median [IQR]	67 [55–76]	64.5 [53–75.7]	73 [67–77]	**<0.001**
Sex: M/F (%)	210/86 (70.1/29.0)	156/74 (67.8/32.2)	54/12 (81.8/18.2)	**0.027**
Oxygen supply/ventilation support AA/VMK/NRM/NIV/OTI (%)	59/74/16/99/48 (20.0/25.0/5.4/33.4/16.2)	59/72/9/84/6 (25.6/31.3/3.9/36.5/2.6)	0/2/7/15/42 (0/3/10.6/22.7/63.6)	**<0.001**
ICU admission (%)	63 (21.3)	20 (8.7)	43 (65.2)	**<0.001**
Time from symptoms’ onset to 1st NPh-S median days [IQR]	7 [3–10]	7 [4–10]	5 [2–9]	0.075
Time from symptoms’ onset to 1st TBNK median days [IQR]	9 [5–12]	9 [6–12]	8 [4–11]	0.089
Comorbidities:				
N# Comorbidity [IQR]	2 [1–3]	2 [1–3]	2 [2–4]	**<0.001**
Any (%)	251 (84.8)	187 (81.3)	64 (97.0)	**0.002**
Smoke (%)	27 (9.1)	20 (8.7)	7 (10.6)	0.635
Obesity (%)	59 (19.9)	47 (20.4)	12 (18.2)	0.686
Cardiovascular (%)	186 (62.6)	137 (59.6)	49 (74.2)	**0.03**
Diabetes (%)	80 (26.9)	60 (26.1)	20 (33.3)	0.497
Endocrinologic (%)	30 (10.1)	25 (10.9)	5 (7.6)	0.434
Cerebrovascular (%)	28 (9.4)	14 (6.1)	14 (21.2)	**<0.001**
Hepatitis (%)	6 (2)	4 (1.7)	2 (3)	0.512
Pulmonary (%)	47 (15.8)	28 (12.2)	19 (28.8)	**0.001**
Renal (%)	23 (7.7)	12 (5.2)	11 (16.7)	**0.002**
Solid Tumor (%)	42 (14.1)	27 (11.7)	15 (2.7)	**0.024**
Hematologic (%)	20 (6.7)	12 (5.2)	8 (12.1)	**0.049**
Neurologic/Psychiatric (%)	36 (12.1)	26 (11.3)	10 (15.1)	0.399
Rheumatologic (%)	14 (4.7)	13 (5.6)	1 (1.5)	0.163
Other (%)	49 (16.5)	36 (15.6)	13 (19.7)	0.436

Quantitative variables are presented as median [IQR]; categorical variables are presented as absolute frequency (percentage). Differences between groups were assessed using the Mann–Whitney U test (continuous variable) or the Chi^2^ test (categorical variables), as appropriate. A two-sided *p* value of <0.05 was considered statistically significant. Statistically significant *p* values are highlighted in bold. IQR: interquartile range, AA: ambient air, VMK: Venturi mask, NRM: nonrebreather oxygen mask with concentrator, NIV: noninvasive mechanical ventilation, OTI: Orotracheal Intubation for mechanical ventilation, ICU: intensive care unit, NPh-S: nasopharyngeal swab for SARS-CoV-2 molecular test, TBNK: T-, B-, Natural Killer-lymphocyte subsets assessment; N#: number.

**Table 2 biomedicines-10-02788-t002:** Laboratory parameters of the study cohort, overall and after stratification for outcome (survivors and non-survivors).

Parameter	All Patients	Survivors	Non-Survivors	*p*
	Median [IQR]	Median [IQR]	Median [IQR]	
RBC (10^6^/µL)	4.46 [4.02–4.92]	4.51 [4.16–4.92]	4.26 [3.78–4.96]	0.061
Hb (g/dL)	13.00 [11.70–14.27]	13.00 [11.70–14.22]	12.70 [11.62–14.22]	0.605
Plt (/µL)	226,000 [175,000–304,000]	225,000 [183,000–315,250]	229,000 [137,250–276,000]	0.102
WBC (/µL)	6920 [5168–9680]	6755 [4945–9217]	8530 [5777–11,832]	**0.004**
Neut (/µL)	5680 [3550–8230]	5035 [3152–7710]	7540 [4840–10,710]	**<0.001**
Ly (/µL)	900 [585–1270]	1020 [628–1325]	520 [390–780]	**<0.001**
N/L ratio	6.69 [3.32–12.04]	5.35 [2.44–9.48]	12.19 [7.74–20.95]	**<0.001**
CRP (mg/L)	65.80 [31.85–121.78]	48.70 [25.05–102.75]	118.90 [75.70–150.55]	**<0.001**
LDH (IU/L)	342.00 [252.00–438.00]	313.50 [235.25–397.00]	484.00 [384.00–616.00]	**<0.001**
CPK (IU/L)	81.00 [37.00–155.50]	72.00 [33.00–133.00]	111.00 [56.00–320.00]	**<0.001**
AST (IU/L)	36.00 [26.00–49.00]	34.00 [25.00–44.00]	47.00 [36.50–65.00]	**<0.001**
ALT (IU/L)	32.00 [17.75–49.00]	30.00 [18.00–47.00]	35.00 [16.00–57.25]	0.437
Conj Bil (mg/dL)	0.27 [0.21–0.35]	0.26 [0.20–0.34]	0.29 [0.23–0.40]	**0.024**
Total Bil (mg/dL)	0.55 [0.44–0.75]	0.55 [0.43–0.74]	0.59 [0.45–0.81]	0.397
Creatinine (mg/dL)	0.88 [0.73–1.07]	0.85 [0.73–1.01]	0.97 [0.80–1.38]	**0.005**
IL-6 (pg/mL)	13.60 [6.77–31.40]	11.30 [5.59–21.22]	37.30 [16.20–60.00]	**<0.001**
TNF alpha (pg/mL)	11.34 [6.72–18.01]	11.39 [6.28–17.98]	11.25 [6.78–17.35]	0.921
D-Dimers (ng/mL)	952.00 [541.00–1538.50]	840.00 [498.00–1466.00]	1237.50 [924.75–2028.00]	**<0.001**
Fibrinogen (mg/dL)	561.00 [429.00–687.00]	550.00 [412.00–681.00]	617.50 [486.75–748.00]	**0.044**
INR	1.16 [1.09–1.22]	1.15 [1.09–1.21]	1.20 [1.13–1.30]	**0.008**
PT (s)	13.90 [13.00–14.60]	13.70 [13.00–14.50]	14.30 [13.50–15.47]	**0.008**
PT (%)	82.00 [75.00–90.00]	83.00 [76.00–90.00]	78.00 [69.25–85.00]	**0.009**

Quantitative variables are presented as median [IQR]. Differences between groups were assessed using the Mann–Whitney U test (continuous variable). A two-sided *p* value of <0.05 was considered statistically significant. Statistically significant *p* values are highlighted in bold. IQR: interquartile range; RBC: red blood cell count; Hb: hemoglobin; Plt: platelet count; WBC: white blood cells count; Neut: neutrophil absolute count; Ly: lymphocyte absolute count; N/L ratio: Neutrophil-to-lymphocyte ratio; CRP: C-Reactive Protein; LDH: lactate dehydrogenase; CPK: creatin phosphokinase; AST: aspartate aminotransferase; ALT: alanine aminotransferase; Conj Bil: conjugated bilirubin; Total Bil: total bilirubin; IL-6: Interleukin-6; TNF: tumor necrosis factor; INR: international normalized ratio; PT: prothrombin time. Reference values: RBC (106/µL): 3.5–5.2; Hb (g/dL): 12.00–16.00; Plt (/µL): 150,000–450,000; WBC (/µL): 4300–10,800; CRP (mg/L): 0–5.00; LDH (UI/L): 125.00–220.00; CPK (UI/L): 30.00–200.00; AST (UI/L): 5.00–34.00; ALT (UI/L): 0–55.00; Conj Bil (mg/dL): ≤0.50; Tot Bil (mg/dL): ≤1.20; Creatinine (mg/dL): 0.73–1.18; IL-6 (pg/mL): <50; TNF-alpha (pg/mL): <12.4; D-dimers (ng/mL): 0–500.00; Fibrinogen (mg/dL): 200.00–400.00; INR 0.80–1.20; PT (%): 70.00–130.00.

**Table 3 biomedicines-10-02788-t003:** T-, B-, and NK-lymphocyte subpopulation absolute counts at hospitalization in the study cohort, overall and after stratification for outcome (survivors and non-survivors).

	All Patients	Survivors	Non-Survivors	*p*
	Median [IQR]	Median [IQR]	Median [IQR]	
CD3+ #	586.50 [334.50–936.25]	666.50 [393.25–1021.75]	320.50 [207.75–533.50]	**<0.001**
CD3+CD4+ #	340.00 [192.50–563.50]	397.50 [236.00–627.00]	199.00 [123.25–340.75]	**<0.001**
CD3+CD8+ #	181.50 [104.75–327.50]	220.00 [128.50–366.25]	103.00 [67.75–164.00]	**<0.001**
CD3+CD4+CD8+ #	7.00 [4.00–13.00]	8.00 [5.00–14.00]	3.50 [2.00–7.75]	**<0.001**
CD3+CD4-CD8- #	19.00 [9.00–39.25]	24.50 [11.25–45.00]	8.00 [4.00–18.75]	**<0.001**
CD19+ #	100.50 [55.75–159.25]	110.00 [62.00–166.00]	74.50 [36.00–105.50]	**<0.001**
CD3^neg^CD16+CD56+ #	128.00 [72.50–211.25]	131.50 [80.00–214.50]	112.50 [66.25–183.50]	0.143
CD4/CD8 RATIO	1.88 [1.19–2.75]	1.87 [1.19–2.65]	1.89 [1.37–2.96]	0.321
TLSI	1.00 [0.00–2.00]	1.00 [0.00–2.00]	2.00 [1.00–2.00]	**<0.001**

Quantitative variables are presented as median [IQR]. Differences between groups were assessed using the Mann–Whitney U test (continuous variable). A two-sided *p* value of <0.05 was considered statistically significant. Statistically significant *p* values are highlighted in bold. IQR: interquartile range; #: absolute count; CD3+: total T-lymphocytes; CD3+CD4+: CD4+ T-Lymphocytes; CD3+CD8+: CD8+ T-Lymphocytes; CD3+CD4+CD8+: CD4+CD8+ double positive T-Lymphocytes; CD3+CD4-CD8-: CD4-CD8- double negative T-Lymphocytes; CD19+: B-Lymphocytes; CD3negCD16+CD56+: CD16+CD56+ NK-cells; CD4/CD8 RATIO: CD4-to-CD8 ratio; TLSI: T-lymphocyte subset index, obtained by computing the number of T-lymphocyte subsets under the cutoff value (range: 0–2). Reference values: CD3+ (cells/μL): 690–2540; CD3+CD4+ (cells/μL): 410–1590; CD3+CD8+ (cells/μL): 190–1140; CD19+ (cells/μL): 90–660; CD3negCD16+CD56+ (cells/μL): 90–590; CD4/CD8 ratio: 1.5–2.5. TLSI CD3+CD4+ cutoff: <369 cells/μL; TLSI CD3+CD8+ cutoff: <194 cells/μL.

**Table 4 biomedicines-10-02788-t004:** Multivariable logistic regression analysis for in-hospital 30-day mortality in patients with SARS-CoV-2 infection.

Parameter	Odds Ratio	OR 95% Confidence Interval		*p*
		Lower Bound	Upper Bound	
Sex (M)	1.719	0.612	4.830	0.304
Age	1.022	0.985	1.061	0.241
N# of comorbidities	1.622	1.179	2.232	**0.003**
Delta Symp-TBNK	0.877	0.795	0.966	**0.008**
LDH	1.010	1.006	1.014	**<0.001**
IL-6	1.007	0.998	1.016	0.139
TLSI	1.893	1.053	3.405	**0.033**

Statistically significant *p* values are highlighted in bold. OR: odds ratio; M: male; N#: number; Delta Symp-TBNK: days between symptoms’ onset and peripheral blood T-, B-, NK-cells assessment; LDH: lactate dehydrogenase; IL-6: interleukin-6; TLSI: T-Lymphocyte subset index.

## Data Availability

The data presented in this study are available on request from the corresponding author. The data are not publicly available due to privacy and ethical restrictions.

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
