# Peer review of "Validation of the T-Lymphocyte Subset Index (TLSI) as a Score to Predict Mortality in Unvaccinated Hospitalized COVID-19 Patients"

_biomedicines, 2022, doi:10.3390/biomedicines10112788_

Round 1
Reviewer 1 Report
The topic is interesting and clinically useful.
The tables and the figures are clear to understand.
The citation are fairly up-to-date.
The format and the English can be slightly better.
Author Response
We thank the Reviewer for the positive comments. We thoroughly revised the manuscript for English language, grammar and style.
Reviewer 2 Report
REVIEWER’S COMMENT
1. This article titled “Validation of the T-lymphocyte subset index (TLSI) as a score to predict mortality in unvaccinated hospitalized COVID-19 patients“ is well written and comprehensive.
2. This is a single-center retrospective study, suggest multi-center study.
3. Any relationship of TLSI among vaccinated hospitalized COVID patients
4. Consider add BMI, SMOKING HISTORY, ALCOHOL HISTORY in this study
5. Suggest make a graphical abstract for this study.
6. ENGLISH EDITING IS NEEDED.
Author Response
- We thank the Reviewer for the comments.
- We agree with the Reviewer and we think that a multi-center study could lead to more reliable results and allow for the generalization of the evidences obtained in our research. Nevertheless, in the discussion section we have compared the cutoffs for CD3+CD4+ and CD3+CD8+ T-lymphocyte absolute counts identified in our study with those reported by other authors in different settings, underlining the similarity of these cutoffs. Indeed, a further direction for this research will be represented by a multi-center study to consolidate our results. A sentence in the discussion section highlights this limitation of the study. Line 340: “Our study has some limitations, such as its retrospective and single-center design.”
- In the present study, the retrospective cohort was exclusively composed by patients hospitalized in the Infectious Disease ward of Policlinico Tor Vergata, from September 2nd, 2020 to December 31st, 2020, therefore before the beginning of the Italian SARS-CoV-2 vaccination campaign. None of the patients received SARS-CoV-2 vaccination before hospitalization. We thank the reviewer for the comment, as it would be very interesting to assess the predictive value on mortality of the TLSI in a new cohort of vaccinated hospitalized COVID-19 patients. A sentence was added to the manuscript in the result section to clarify this point. Lines 118-119: “None of the enrolled patients had received SARS-CoV-2 vaccination.”
- Smoking habits and obesity have been already included in the analysis (see table 1), and no differences were identified after comparing Survivors with Nonsurvivors. Moreover, no correlations between TLSI and smoking habits or obesity were identified. One limitation of our study is represented by the fact that obesity was recorded as a categorical variable (considering a BMI cutoff of 30 Kg/m2) and we did not register the exact BMI for each patient. We added a sentence to clarify this point. Lines 276-279: “Moreover, in the present study obesity was recorded as a categorical variable (considering a body mass index [BMI] cutoff of 30 Kg/m2) without recording the exact BMI value for each patient. This could represent a limitation of our analysis”. As for alcohol consumption, all the patients declared no alcohol abuse, therefore the variable was not included in the analysis.
- A graphical abstract was realized and uploaded with the revised manuscript.
- We thoroughly revised the manuscript for English language, grammar and style.
Reviewer 3 Report
In the manuscript " Validation of the T-lymphocyte subset index (TLSI) as a score to predict mortality in unvaccinated hospitalized COVID-19 patients ", it was reported that lymphopenia was correlated with severe Coronavirus Disease-2019 (COVID-19).
The authors evaluated the role of T-lymphocyte subset absolute counts measured at ward admission in predicting the 30-day mortality in COVID‐19 hospitalized patients, applying a new prognostic score based on the T-Lymphocyte Subset Index (TLSI). They demonstrated that TLSI represents a valuable tool for rapid and accurate mortality risk stratification in COVID-19 patients.
The work is very interesting and innovative. The results are explained very well, and the materials and methods are described in detail.
Author Response
We thank the reviewer for the positive comments.